# In Vitro and In Vivo Efficacy of Topical Dosage Forms Containing Self-Nanoemulsifying Drug Delivery System Loaded with Curcumin

**DOI:** 10.3390/pharmaceutics15082054

**Published:** 2023-07-31

**Authors:** Gréta Frei, Ádám Haimhoffer, Enikő Csapó, Krisztina Bodnár, Gábor Vasvári, Dániel Nemes, István Lekli, Alexandra Gyöngyösi, Ildikó Bácskay, Pálma Fehér, Liza Józsa

**Affiliations:** 1Department of Pharmaceutical Technology, Faculty of Pharmacy, University of Debrecen, 4032 Debrecen, Hungary; freigreta07@gmail.com (G.F.); haimhoffer.adam@euipar.unideb.hu (Á.H.); csapoeniko0716@gmail.com (E.C.); b.kriszti1999@gmail.com (K.B.); vasvari.gabor@pharm.unideb.hu (G.V.); nemes.daniel@pharm.unideb.hu (D.N.); bacskay.ildiko@pharm.unideb.hu (I.B.); feher.palma@pharm.unideb.hu (P.F.); 2Healthcare Industry Institute, University of Debrecen, 4032 Debrecen, Hungary; lekli.istvan@pharm.unideb.hu (I.L.); gyongyosi.alexandra@pharm.unideb.hu (A.G.); 3Department of Pharmacology, Faculty of Pharmacy, University of Debrecen, 4032 Debrecen, Hungary

**Keywords:** curcumin, self-nanoemulsifying drug delivery systems, drug delivery, anti-inflammatory effect

## Abstract

The external use of curcumin is rare, although it can be a valuable active ingredient in the treatment of certain inflammatory diseases. The aim of our experimental work was to formulate topical dosage forms containing curcumin for the treatment of atopic dermatitis. Curcumin has extremely poor solubility and bioavailability, so we have tried to increase it with the usage of self-emulsifying drug delivery systems. Creams and gels were formulated using penetration-enhancing surfactants and gelling agents. The release of the drug from the vehicle and its penetration through the membrane were determined using a Franz diffusion cell. An MTT cytotoxicity and in vitro antioxidant assays were performed on HaCaT cell line. The in vivo anti-inflammatory effect of the preparations was tested by measuring rat paw edema. In addition, we examined the degree of inflammation induced by UV radiation after pretreatment with the cream and the gel on rats. For the gels containing SNEDDS, the highest penetration was measured after half an hour, while for the cream, it took one hour to reach the maximum concentration. The gel containing Pemulen TR-1 showed the highest drug release. It was determined that the curcumin-containing preparations can be safely applied on the skin and have antioxidant effects. The animal experiments have proven the effectiveness of curcumin-containing topical preparations.

## 1. Introduction

The skin acts as a protective barrier against mechanical and physical injuries, pathogens, and chemicals. However, exposure to ultraviolet (UV) radiation can already lead to skin damage and the increased production of reactive oxygen species (ROS), resulting in inflammation, wrinkles, dryness, roughness, reduced elasticity, or in severe cases, skin cancer [1,2]. UV radiation, especially UVB, can damage the deoxyribonucleic acid (DNA) and protein structures of the epidermis, which is the outermost part of the skin and induces local inflammation. The long-term accumulation of inflammatory factors, thereby maintaining the inflammatory microenvironment in the skin tissue, increases the risk of skin cancer and metastases [2,3,4].

Nowadays, more and more people prefer the use of medicines and medicinal products containing active ingredients of natural origin, both in the treatment of diseases and for preventive purposes. According to the data from the World Health Organization (WHO), at least 80 percent of the world’s population has already been treated with active ingredients of natural origin or used natural remedies as complementary therapy [5]. These are characterized by a more favorable patient compliance and side effect profile. In the case of topical formulations, many types of plant extracts are used, with a variety of chemical ingredients, for example, flavonoids, tannins, phenolics, aminoacids, and vitamins, which influence the biological functions of the skin [6,7]. However, the low bioavailability and instability of these natural products may raise concerns. To fix the mentioned problems, a proper drug delivery system is needed to enhance their efficacy and stability [8].

To the best of our knowledge, there is no treatment for inflammatory skin diseases that results in a complete cure. The use of steroid-containing preparations reduces inflammatory complaints, but their long-term application is not recommended, as they have many side effects, such as skin atrophy, hyper- and hypopigmentation, and skin burns. [9]. Therefore, there is a need for an alternative treatment for dermatitis with fewer effects.

Turmeric (*Curcuma longa*, *C. longa*) has been used in both the cosmetic and pharmaceutical industries. The polyphenolic compound called curcumin—which is the main active ingredient of the plant—can be isolated from its rhizome [10].

Based on the literature, curcumin and its derivates can be promising in the treatment of many dermatological diseases [11] thanks to their significant anti-inflammatory and antioxidant activities. According to the literature, curcumin can be effective in the treatment of several inflammatory skin diseases, including acne, atopic dermatitis, psoriasis, and vitiligo [12]. Numerous clinical trials have shown statistically significant improvements in inflammatory skin disease severity in the curcumin-treated groups compared to the control groups [13,14,15]. Based on these above-mentioned findings, there is early evidence that topical formulations containing curcumin may have therapeutic benefits; however, further investigations are needed to evaluate its efficiency and the mechanisms of action [12]. According to several scientific studies, curcumin reduces the production of free radicals and inflammation through nuclear factor-κB (NF-κB) inhibition [16]. It can directly bind to the receptor-binding sites of tumor necrosis factor alpha (TNF-α), blocking the subsequent TNF-dependent activation of NF-κB [17,18]. On the other hand, in the cosmetic industry, curcumin is often used because of its ultraviolet B (UVB) protective effect. [19].

However, curcumin has extremely poor bioavailability, which may result in reduced efficacy. According to the Biopharmaceutical Classification System (BCS), curcumin belongs to the group IV, which means that it shows very low water-solubility and permeability [20]. Therefore, the in vivo effectiveness of curcumin is usually limited in aqueous solution. The highly hydrophobic properties of curcumin combined with the barrier function of the skin result in very low percutaneous penetration [21] However, when it is incorporated into a drug delivery system such as chitosan-alginate microcapsules, nano-emulsion, or combined with β-cyclodextrin to form a nanoparticle complex, it resulted in good bioavailability and bioactivity [18,22,23,24]. Thanks to these novel drug delivery systems, curcumin is able to penetrate into the dermis and has become suitable as a therapeutic agent for the local treatment of various inflammatory skin conditions. According to the literature, nanocarrier systems make it possible to enhance the permeation of curcumin through the stratum corneum. Ex vivo skin permeation studies have confirmed that the nanoparticles are able to interact with the stratum corneum cells, thus enhancing permeation and accumulation in the epidermal and dermal region [21]. Moreover, permeation of curcumin is found to increase 7–8-fold when formulated with permeation enhancers and in the form of a gel [25,26].

This study was aimed to design topical dosage forms containing curcumin in self-nanoemulsifying drug delivery system and to evaluate their in vitro and in vivo UV protective, anti-inflammatory, and antioxidant effects. In our previous study it has been proved that the incorporation of curcumin into self-nano- and microemulsifying drug delivery systems significantly improved the solubility and the bioavailability of the active ingredient, and thus the antioxidant and anti-inflammatory effect were also increased [27]. Therefore, the curcumin containing topical formulations introduced in our present article were designed as described above: curcumin incorporated into the mentioned self-nanoemulsifying drug delivery system (SNEDDS) was used during the formulation of the novel cream and gel preparations. The consistency, the viscosity, and the spreadability of the formulations were investigated. In vitro permeation assays were conducted to compare the permeated amount of curcumin from the topical formulations with and without SNEDDS. In order to determine the kinetic profile of the curcumin, the diffusion coefficients and release rates were measured. In vitro cell viability tests were performed on HaCaT cells before and after UVB irradiation with the help of 3-(4,5-dimethylthiazol-2-yl)-2,5-diphenyl tetrazolium bromide (MTT) dye. Different in vitro antioxidant capacity assays (2,2-diphenyl-1-picrylhydrazyl (DPPH) and superoxide dismutase (SOD) assay) were carried out to evaluate the direct and indirect antioxidant activity. The in vivo anti-inflammatory effect of the selected preparations was investigated by measuring rat paw edema. Finally, TNF-α and interleukin-1 beta (IL-1β) enzyme levels were also evaluated in the skin tissue of premedicated rats after UVB irradiation.

## 2. Materials and Methods

### 2.1. Materials

Labrasol (caprylocaproyl polyoxyl-8 glyceride), Tefose 63 (mixture of PEG-6 stearate, ethylene glycol palmitostearate and PEG-32 stearate), and Sedefos 75 (mixture of triceteareth-4 phosphate, ethylene and diethylene glycol stearate) were obtained from Gattefossé (Lyon, France). Cremophor RH 40 (polyoxyl 40 hydrogenated castor oil) was obtained from BASF Company (Ludwigshafen, Germany). Pemulen TR-1 and Carbopol 974P were purchased from Lubrizol Corporation, Ohio, USA. Stearic acid, cetylstearyl alcohol, propylene glycol, and isopropyl myristate were purchased from Hungaropharma Ltd., (Budapest, Hungary). The MTT dye, curcumin, DPPH free radical, human and rat TNF-α and IL-1β ELISA Kits and ingredients needed for the cell maintenance were purchased from Sigma-Aldrich (Budapest, Hungary). The immortalized human keratinocyte cell line (HaCaT) was obtained from Cell Lines Service (CLS, Heidelberg, Germany).

### 2.2. Formulation of Self-Nanoemulsifying Drug Delivery Systems

The extremely low water solubility of curcumin necessitated the use of an innovative drug carrier system for the formulation. Therefore, we used a self-nanoemulsifying drug delivery system (SNEDDS) developed and published by us earlier [27]. According to our previous results, the applied drug carrier system can greatly increase the solubility and thus the bioavailability of curcumin. SNEDDS have been prepared by mixing of a surfactant (Labrasol) and a co-surfactant (Transcutol P) in 1:1 ratio. The given concentration of curcumin was dissolved in the mixture of tensides at 24.5 °C by permanent agitation. The composition of the SNEDDS used is presented in Table 1. According to our previous study, the droplet size of this composition was 68.45 ± 3.66 nm, while the polydispersity index (PDI) was 0.186 ± 0.014, which indicated a homogenous size distribution [27].

### 2.3. Formulation of Topical Dosage Forms

Curcumin-containing creams and gels were formulated using two different emulsifiers (Tefose 63 and Sedefos 75) and two different gelling agents (Pemulen TR-1 and Carbopol 974P) as well. Topical dosage forms are created also with and without SNEDDS.

Table 2 shows the compositions of the formulated creams. The creams were prepared in two steps, thus creating an emulsion system. The lipophilic phase was formulated by melting the cetylstearyl alcohol, stearic acid, and IPM at 80 °C. The aqueous solution of Tefose 63 or Sedefos 75 surfactants and propylene glycol was added to the oily phase with continuous stirring at 400 rpm by Radelkis OP-912 magnetic stirrer (Radelkis, Budapest, Hungary).

Table 3 presents the compositions of the curcumin-containing gels. In the case of the formulation of the gels, an appropriate amount of polymer (Pemulen TR-1 or Carbopol 974P) was added to the indicated amount of water, and with the addition of triethanolamine, the gelation process was started. Gels signed with V and VII contained Carbopol 974P polymer, while VI and VIII were made with Pemulen TR-1 gelling agent. Curcumin was suspended in the prepared gels (gel compositions V and VI) or distributed as self-nanoemulsifying drug delivery systems (gel compositions VII and VIII).

### 2.4. Texture Analysis of Creams and Gels

The investigation of the textural properties of the formulations was performed with a CT3 Texture Analyzer (Brookfield, Middleboro, MA, USA). This device is capable of measuring and calculating physical properties that have been shown to be closely related to human sensory evaluation of topical formulations. A tension and a compression test were also performed. The instrument was equipped with a TA5 Cylinder type probe (35 mm length and 12.7 mm diameter). The trigger load was 3.0 g, while the speed was set to 0.45 mm/s.

The amount of force required to penetrate and remove the probe from the creams and gels was detected with Texture Pro CT Software version 1.3 (Brookfield Engineering Laboratories, Middleborough, MA, USA). During the compression measurement, the probe was touched directly to the surface of the sample to be examined, and then the measurement was started. In the case of the tension test, the probe was placed in the cream or gel (up to 5 mm), and thus the magnitude of the force required for lifting was examined. The compression and the tension studies were performed at room temperature (24 ± 1 °C).

### 2.5. Viscosity and Spreadability

The viscosity measurements of the compositions with and without curcumin were performed with RheolabQC Rotational Rheometer. Half gram of the preparations was added to the cup of the concentric cylinder measuring system (d = 26.7 mm). The viscosity curves of the creams and gels were determined by rotation tests at a controlled shear rate ranging from 2.0 to 50.0 s^−1^ at room temperature.

The spreadability of the formulations was calculated in terms of the diameter of the circle produced by the creams and gels when placed between two glass plates at room temperature. Before the test, a circle with a diameter of 2 cm was marked on a glass plate. One gram of the cream or gel was placed within the circle, then a second plate with a weight of 200 g was placed over it. The slides were allowed to remain in place for 5 min. The increase in the diameter of the circle was determined [28].

### 2.6. In Vitro Permeation Studies

The in vitro permeation studies were carried out with a Franz diffusion cells (Hanson Microette^TM^ Topical and Transdermal Diffusion Cell System).

For the test, a lipophilic surface was created similar to the skin: cellulose acetate synthetic membranes were soaked in isopropyl myristate for 10 min before the studies. This synthetic membrane provides an opportunity to mimic human skin and to monitor the permeation of the active ingredient (curcumin) through the skin. The diffusion surface area of the membranes was 1.767 cm^2^. The receptor medium (7.0 mL) was obtained by mixing 1:1 ethanol (70 *v/v*%) and pH 7.4 phosphate-buffered saline (PBS) in order to maintain sink conditions [29]. The rotation speed of the magnetic stirrer was 350 rpm, while the temperature of the receptor phase was 32 ± 0.5 °C.

A quantity of 0.5 g from each composition was placed on the membrane (equivalent to 20 mg of curcumin), and the cells were closed with a glass disc and a retaining ring.

The in vitro permeation studies were performed for 240 min. Samples of 1.0 mL were taken at the given times and replaced with the same amount of fresh receptor phase. The absorbance of the samples was measured with UV spectrophotometer (Shimadzu, Tokyo, Japan) at 425 nm, and the curcumin content of the samples was determined [30]. The mixture of ethanol (70 *v*/*v*%) and PBS (1:1) was used as a blank sample. A calibration curve was determined before the spectroscopic measurements of curcumin.

To compare the permeation profiles of the cream/gel compositions with or without SNEDDS, difference factor (*f*_1_) and similarity factor (*f*_2_) were calculated. Difference factor (*f*_1_) is the percentage difference between two curves at each point, while the similarity factor (*f*_2_) is the similarity between the two curves. The following equations (Equations (1) and (2)) were used to calculate *f*_1_ and *f*_2_ values:

(1)$$f_{1}=[\sum (R_{t}-T_{t})/\sum R_{t}]\times 100$$(2)$$f_{2}=50\times log[\{1+(R_{t}-T_{t})\times 1/n\}-0.5]$$
where *n* is the sampling number, *R_t_* and *T_t_* are the percentages of dissolved reference and test products at each time point *t*. Two dissolution profiles are considered similar and bioequivalent if *f*_1_ is between 0 and 15 and *f*_2_ is between 50 and 100 [31].

The curcumin release rate (k) was calculated from the slope of the amount of drug released per unit area (µg/cm^2^) plotted as a function of the square root of time (min½). The diffusion coefficient (*D*) was determined from the drug concentration at a given t time (*Q*, µg/cm^2^), the initial concentration ($C_{0}^{\prime }$), and the diffusion time (*t*) (Equation (3)) [31]:(3)$$D=\frac{Q^{2}\times \pi }{{\left(2[C_{0}^{\prime }]\right)}^{2}\times t.}$$

### 2.7. In Vitro Cell Viability before and after UV-B Radiation

For the cell viability assay, test solutions were prepared from creams and gels using sterile pH 7.4 PBS solution. One gram of the formulations was dispersed in 100–100 mL PBS, and they were mixed for 3 h by Radelkis OP-912 magnetic stirrer (Radelkis, Budapest, Hungary) at room temperature. Compositions with or without curcumin was also investigated.

The viability of the HaCaT cells was determined by MTT assay according to the method described by Mosmann [32]. The cells were maintained by weekly passages in DMEM culture media and seeded into 96-well plates at a density of 10^4^ cells/well. After 5 days, when 80% confluence was achieved, the medium was removed, and the cells were treated with 100 µL of the samples for 1 h at 37 °C. At the beginning of the treatment, certain wells were exposed to UV-B radiation for 10 min with a dose 30 mJ/cm^2^ [33]. At the end of the incubation time, the samples were removed, and 0.5 mg/mL MTT solution (dissolved in pH 7.4 PBS) was added for 3 h. After removing the MTT solution, acidic isopropanol (isopropanol:1.00 N hydrochloric acid = 25:1) was added to dissolve the created formazan precipitate. Finally, the absorbance of the solutions was detected at 565 nm. PBS and Triton × 100 (10% *w*/*v*) solutions were used as negative and positive controls, respectively. The results were demonstrated as a percentage of the viability of the control cells, incubated for 1 h only with PBS, and were not exposed to UV radiation.

### 2.8. Determination of Superoxide Dismutase Enzyme Activity on HaCaT Cells after UV-B Radiation

For the investigation, UV-B (Oriel^®^ Sol-UV-4 UV Solar Simulator, Bullville, NY, USA) radiation was used to induce oxidative stress before or after the treatment. The antioxidant activity of the formulations was determined before and after UV-B exposure. HaCaT cells were seeded into a 12-well plate (10^5^ cells/well). When the cells fully grow over the wells’ membrane, the medium was fully removed, and 200 µL of the samples were added to each well. The cells were pre-and post-treated with the samples and incubated for 30 min at 37 °C in an atmosphere of 5% CO_2_. Trolox (10 µM) and PBS were used as positive and negative controls, respectively. The formulated creams and gels were investigated in a concentration of 5 *w*/*v*%, dissolved in PBS. The cultured HaCaT keratinocytes were irradiated with 60 mJ/cm^2^ UV-B radiation for 5 min, and the dose was determined to be sufficient to induce oxidative stress. After the 30 min incubation time, the samples were removed, and then the cells were collected and centrifuged with 20 mM HEPES buffer (pH 7.2) (3500 rpm, 5 min 4 °C).

The antioxidant enzyme (SOD) activity was determined using Cayman kit (Cayman Chemical, Ann Arbor, MI, USA) according to the instructions of the manufacturer. Ten microliters of the supernatant were incubated with the detection solution for 2 min at room temperature. Then, 20 µL xanthine oxidase enzyme was added to each well. Finally, the absorbance was measured at 450 nm after 30 min incubation at room temperature. SOD activity was expressed in U/mL, which corresponds to the amount of SOD that inhibits tetrazolium salt reduction by 50%.

### 2.9. Direct Antioxidant Capacity Test

The direct radical scavenging activity of the samples was determined using 2,2-diphenyl-1-picrylhydrazyl (DPPH) free radical (Brand–Williams method). For the experiment, DPPH was dissolved in absolute ethanol (0.06 mM). The reaction mixture was created by mixing 2 mL of DPPH solution and 900 µL of absolute ethanol. One hundred microliters of the samples was added to the reaction mixture and incubated for 30 min. The reaction of DPPH with free radicals resulted in a color change in the solution [34]. The amount of free DPPH—which did not react with antioxidants—was measured with UV spectrophotometer at 517 nm [35]. Ascorbic acid (0.25 mg/mL) served as positive control [36], while DPPH solution was used as negative control. The percentage of the inhibited reactive oxygen species (AA%—antioxidant activity) was determined with the following equation (Equation (4)) [37]:AA% = 100 − {[(Abs_sample_ − Abs_blank_) × 100]/Abs_control_}(4)

### 2.10. Examination of In Vitro Anti-Inflammatory Effect

To investigate the in vitro anti-inflammatory effect of the topical formulations human IL-1β (Sigma—RAB0273) and human TNF-α (Sigma—RAB0476), ELISA tests were carried out on HaCaT cell line in 96-well plates (10^4^ cells/well) according to the manufacturer’s instructions. Test solutions were created by mixing the formulated creams and gels with PBS in a concentration of 3 *w*/*w*%. From each sample, 200–200 μL was added and incubated for 1 h. At the end of the incubation time, the test solutions were removed, and 50 μL of IL-6 (20 ng/mL) was added to each well and incubated for 16 h at 37 °C in an atmosphere of 5% CO_2_. The next day, the supernatant was removed, and the above-mentioned human ELISA tests were performed.

### 2.11. Experimental Animals

Male Rattus Norvegicus (SPRD) rats (Charles River Laboratories International, Inc., Sulzfeld, Germany) with an average weight of 526 ± 42 g were used for the in vivo experiments. The rats were kept and treated according to the “Principles of Laboratory Animal Care” formulated by the National Society for Medical Research and the “Guide for the Care and Use of Laboratory Animals” prepared by the National Academy of Sciences and published by the National Institutes of Health (NIH Publication no. 86–23, revised in 1996). Animals were housed in wire-bottomed cages maintained on 12:12 h light–dark cycle throughout the study and nutrified with standard rodent chew pellets ad libitum with free access to water. Approval number: 10/2022/DEMÁB.

### 2.12. Carrageenan-Induced Acute Inflammatory Model

Anti-inflammatory activity was measured using carrageenan-induced rat paw edema assay [38,39]. For the in vivo experiments, the rats were anesthetized with ketamine/xylazine (50/10 mg/kg body weight). After that, the thickness of the rats’ paws was measured with an engineering precision caliper, and then the outer and inner surface of their left front paw was pretreated with the selected cream and gel formulations (Sedefos + SNEDDS and Pemulen + SNEDDS), which showed the best results in the in vitro experiments or with a cream/gel containing diclofenac as positive control. Moreover, the same formulations, but without active ingredients, were also investigated as negative control.

Edema was induced by a subplantar injection of 100 μL of carrageenan solution (1%) into the left front paws. Animals were treated with 1–1 g of cream and gel, respectively, 1 h before the injection. Paw thickness was measured just before the injection (“0 h”) and 4 h after. Increase in paw thickness was measured as the difference in paw thickness at “0 h” and paw thickness at the fourth hour.

### 2.13. Inflammation Induced by UV Radiation after Pretreatment

In the second in vivo experiment, the rats were also treated under anesthesia with ketamine/xylazine (50/10 mg/kg body weight). The animals are depilated, and the tests are performed on the second day after depilation. Animals were divided into two groups: Group A was treated with the gel formulations (with and without the active ingredient) and with the controls, while Group B was treated with the creams (with and without the active ingredient) and with the controls. Under anesthesia, five rectangles of 5 × 1.5 cm were marked on the animals’ skin, and the rest were covered with aluminum foil so that the UV radiation only reaches the treated parts. The samples (0.3 g) were applied to the rectangles in the following order: gel/cream without active ingredient, gel/cream containing 2% active ingredient (Pemulen + SNEDDS/Sedefos + SNEDDS), diclofenac (2%) gel/cream as positive control, and an original cream contain sun protection factors (SPF) (NIVEA Sun Lotion, SPF 30). Thirty minutes after the pretreatment, the animals were placed under a device capable of emitting UV radiation (Oriel^®^ Sol-UV-4 UV Solar Simulator, Bullville, NY, USA). The rats were irradiated for 30 min (UVA + UVB radiation) with a dose of 50 mJ/cm^2^. Six hours after irradiation, the animals were euthanized i.p. with pentobatbital injection (100 mg/kg body weight). The irradiated skin surface was isolated, and the tissue lysates were homogenized 500 μL sterile saline and centrifuged (2000× *g*, 15 min, 4 °C). The supernatants were used to determine the amount of the proinflammatory cytokine levels. For the determination TNF-α and IL-1β ELISA kits (Sigma Aldrich Kft., Budapest, Hungary) were used according to the manufacturer’s instructions.

### 2.14. Statistical Analysis

All data were handled and analyzed by Microsoft Excel 2016 and GraphPad Prism 6. Comparison of the results of the texture analysis, permeation studies, in vitro cell viability, antioxidant and anti-inflammatory tests, and the results of the in vivo experiments was performed with one-way ANOVA followed by Dunnett’s or Tukey’s multiple comparison test [40,41]. Difference of means was considered as statistically significant at *p* < 0.05.

## 3. Results

### 3.1. Macroscopic Properties and pH Measurement

The evaluation of the macroscopic properties of the formulations was performed immediately and 30 and 60 days after preparation. Creams and gels containing SNEDDS had a homogeneous, orange appearance. The formulations without SNEDDS were heterogeneous. The pH value of the compositions was between 5.72 and 6.03 immediately after formulation (Table 4). The pH of the aqueous solution of the used polymers (Carbopol, Pemulen) was between 2.5 and 3.5, depending on the concentration of the polymer, so it was necessary to add triethanolamine (trolamine) to increase the pH [42]. For gels, the optimum viscosity is achieved in the pH range of 5.5–7.0.

As Table 4 shows, there were no changes in pH values of the preparations even after 60 days of storage at room temperature.

### 3.2. Texture Analysis

The compression and tension forces (N) required to insert the probe to a given distance into the formulations or to remove the probe from the formulations are shown in Figure 1a,b. The maximum force values measured in case of the compression and the tensions test indicated the firmness of the formulations [43]. It was found that creams containing curcumin in suspended form (without SNEDDS) demonstrated significantly higher values in comparison with the compositions where curcumin was in SNEDDS. The most significant difference (*p* < 0.001) was observed in the case of the composition containing Tefose emulsifier. In case of the gels, lower resistance was detected both in the compression and tension test, which indicates a lower level of firmness. Conversely, higher values indicate a hard consistency, which may hinder the release of curcumin, thereby decreasing the bioavailability.

### 3.3. Viscosity and Spreadability

The determination of the viscosity provides information on the flow behavior as well as the spreadability of the topical preparations. The viscosity was decreased due to the increasing shear rate in all cases. It can be concluded that each cream and gel are non-Newtonian, shear-thinning formulations, displaying varying degrees of shear thinning. As shown in the Figure 2a,b, there were no significant differences between the same creams/gels with or without curcumin. However, formulations containing curcumin incorporated into SNEDDS are proved to have lower viscosity than the same formulations but without SNEDDS. The lower viscosity suggests better spreadability, which can be beneficial from the point of view of usability.

Although the spreadability of a topical dosage form can be predict from the level of viscosity, another test was also carried out. The results of the spreadability measurements were correlated with the results of the viscosity test. Gels were significantly more spreadable than cream. Furthermore, comparing formulations with and without SNEDDS, it was found that the application of SNEDDS increased the level of spreadability (Table 5).

### 3.4. In Vitro Permeation Studies

In vitro permeation profiles of the cream and gel formulations were investigated with Franz diffusion method. Figure 3a shows the results of the in vitro permeation profiles of curcumin in case of the gels, while Figure 3b represents the permeation profiles from the creams across isopropyl myristate (IPM) impregnated cellulose acetate membrane. The average cumulative percentage of curcumin that permeated through the membrane (%) was plotted against time (minute) in the graphs.

According to our permeation studies, it can be concluded that after one hour, the dissolution curve reaches its maximum in almost all cases, and after that, no significant increase is expected. An exception to this is the product formulated with Carbopol, because, in this case, the maximum permeation can be observed after 2 h.

It can be concluded that the permeation profile of curcumin showed dependence on the type of cream and gel base used. Comparing the penetration profiles from SNEDDS containing creams and gels, higher curcumin concentration was detected regarding the gels. The composition formulated with Pemulen gel forming polymer was the most suitable composition, as 39.86 ± 0.82% of the curcumin was permeated across the membrane after 4 h. Moreover, the release rate was also the highest for this formulation (50.683 ± 3.023 µg/cm^2^ × min½), as it can be seen in Table 6.

When the diffusion coefficients were calculated, it was found that the composition formulated with Pemulen + SNEDDS showed the fastest curcumin diffusion with the value of 0.20479 ± 0.01543 cm^2^/min, while for the same composition but without SNEDDS, the average diffusion coefficient value was only 0.01593 ± 0.00112 cm^2^/min. As Figure 3a,b also show, the release rate and the diffusion coefficient related to the formulations without SNEDDS were very similar after 4 h.

The diffusion profiles of the compositions formulated with the same emulsifying agents but with or without SNEDDS were compared with each other. The calculated difference factors (*f*_1_) and similarity factors (*f*_2_) are shown in Table 7. According to our results, the diffusion profiles of curcumin from composition with and without SNEDDS differed significantly in each case because all difference factors were higher than 15 and all similarity factors were below 50.

### 3.5. In Vitro Cell Viability

MTT cytotoxicity tests were carried out on HaCaT cell monolayer before and after UV-B exposure. Figure 4a,b represent the cell viability before and after UV-B irradiation, respectively, expressed as a percent of the viability of cells treated only with PBS and not exposed to UV radiation.

According to our examination, creams demonstrated higher cell viability values than gels before the irradiation. Regarding the creams, the formulation containing the Sedefos surfactant was less toxic, as the cell viability was 98.2 ± 1.3% in this case. As it can be seen in Figure 4a, the presence of the active ingredient did not influence the level of the toxicity in the case of the non-UV groups.

After UV-B exposure, cells treated with formulations without SNEDDS showed significant decrease of viability. Significant differences were obtained between the formulations with and without curcumin in every case in the UV-exposed group. Based on the results, it was found that treatment with creams and gels containing curcumin incorporated to SNEDDS resulted in higher cell viability. Overall, it can be concluded that treatment with formulations containing curcumin in SNEDDS can prevent the cell-damaging effect of UV radiation.

### 3.6. In Vitro Antioxidant Activity Tests

#### 3.6.1. Superoxide Dismutase Activity on HaCaT Cells

The SOD enzyme activity of the treated groups was compared with the SOD activity of the cells treated with PBS. As positive control the SOD activity of cells which were not exposed to UVB radiation was also measured (no UV). The enzyme activity of cells which were pre- and post-treated with sunscreen (SPF 30) was also examined.

Figure 5a,b show that in irradiated cells, where no pre-treatment or post-treatment of cream/gel was used (treated only with PBS), the SOD activity was significantly decreased in comparison with the non-UV group. In case of the pre-treatment we observed a smaller decrease in SOD values (Figure 5a). The best result was obtained with the pre-treatment of composition containing Pemulen and SNEDDS. In this case, the average SOD activity was 0.822 U/mL after UVB irradiation. The post-treatment could not prevent a significant decrease in the enzyme level caused by UVB radiation. However, treatment with the formulations containing SNEDDS resulted in a higher level of enzyme activity in both cases.

#### 3.6.2. Direct Radical Scavenging Activity

The 2,2-diphenyl-1-picrylhydrazyl (DPPH) method was used to determine free radical scavenging activity (% inhibited reactive oxygen species (ROS)) of the formulations [44], which was calculated for each cream and gel with or without curcumin. Significant differences were detected in every case between the formulations with and without curcumin, except for Tefose-containing cream (Figure 6). It was obtained that the vehicles alone did not show significant antioxidant effect. Based on the results of the test, it was determined that the antioxidant effect was the highest in the case of Pemulen- and SNEDDS-containing gel, as the radical scavenging activity was 32.87 ± 2.05% for this composition. According to the measurements, the application of SNEDDS increased the radical scavenging activity in every cases.

### 3.7. In Vitro Anti-Inflammatory Effect

The anti-inflammatory activity of the different formulations was evaluated by ELISA assays on HaCaT cells. PBS was used as the negative control and taken as 100% during the experiment. The values were expressed as the percentage of the negative control. As it was presented in Figure 7a,b, treatment with the compositions without curcumin did not significantly reduce TNF-α or IL-1β production in the cells.

According to the results of the study, samples containing curcumin incorporated in SNEDDS reduced both proinflammatory cytokines approximately equally. A more significant decrease in TNF-α level was obtained in the case of the preparations containing Pemulen + SNEDDS and Sedefos + SNEDDS. The preparation containing Pemulen + SNEDDS showed the most significant anti-inflammatory effect on HaCaT cells; in this case, both TNF-α and IL-1β levels were the lowest. There were significant differences between the TNF-α and IL-1β levels of the cells which were treated with PBS and with the curcumin + SNEDDS containing formulations. It can also be concluded that the presence of curcumin has improved the anti-inflammatory activity in every case.

### 3.8. In Vivo Experiments

According to the results of the above discussed preformulation and in vitro experiments, the SNEDDS-containing gel with Pemulen base and the SNEDDS-containing cream with Sedefos was selected for further in vivo experiments. Moreover, these formulations were also tested without active ingredient and were regarded as negative controls. As positive control, diclofenac-containing gel and cream were formulated with the same excipients.

#### 3.8.1. Rat Paw Edema Test

Injection of carrageenan into the left front paw induced edema in each case (Figure 8). However, in the case of the pretreatment with gel and cream containing curcumin incorporated to SNEDDS, significantly lower values were detected compared to the other groups. According to the results, it can be concluded that curcumin + SNEDDS-containing preparations were significantly more effective against the carrageenan induced rat paw edema than the diclofenac containing formulations, as they significantly reduced the degree of paw edema 4 h after carrageenan injection.

#### 3.8.2. Effect of the Formulations upon UVB-Induced Skin Inflammation

In the experiments, negative and positive control groups were used to confirm the induction of response to UVB radiation. The unloaded topical formulations were also tested. According to the results, gel and cream formulations containing curcumin incorporated into SNEDDS showed significantly lower cytokine levels in both cases (Figure 9). Formulations without the active ingredient did not influence the production of either TNF-α or IL-1β. Comparing the levels of TNF-α and IL-1β, we found that the amount of TNF-α decreased more significantly as a result of the treatment with the curcumin + SNEDDS-containing gel and cream. Furthermore, our formulations were able to exceed the anti-inflammatory effect of the diclofenac-containing cream and gel in the case of UV-induced inflammation.

## 4. Discussion

The present study describes the development of novel curcumin-containing topical dosage forms and the investigation of their antioxidant and anti-inflammatory potential and UV-protective effect. As it has been previously described, the topically applied products containing natural active substances have an increasing role in the treatment of many inflammatory skin diseases. The external use of curcumin is not so widespread, although it is a valuable active ingredient with antioxidant and anti-inflammatory potential [45,46,47]. Based on the literature, topical use of curcumin has wound healing effects by enhancing re-epithelialization and it can decrease the amount of reactive oxygen species [23,48]. Furthermore, the topical use of curcumin has been reported to protect against UV radiation-induced skin dermatitis. However, the poor solubility and thus the unfavorable bioavailability of curcumin make it difficult to formulate a product with adequate efficiency [18]. To solve this problem, in our previous research, curcumin-containing SNEDDS was formulated. In this study, it has been proven that the incorporation of curcumin into SNEDDS is a promising approach to overcome solubility and bioavailability barriers [27].

The primary objective of our experimental work was to develop novel topical dosage forms (creams and gels) containing curcumin in suspended form or incorporated into SNEDDS and to investigate their efficiency against free radicals, inflammation, and UVB irradiation. For the formulation of gels, Pemulen TR-1 or Carbopol 974P were used as gel forming agents, while for the creams, Tefose 69 or Sedefos 75 nonionic emulgents were chosen. These excipients have been proven to be non-toxic and safe to use in topical preparations [49].

As Bolla et al. described, the textural properties are important factors for topical products since they are able to modify the release and the diffusion rate of the active substances. In the case of our gel formulated with Pemulen, the measured higher diffusion coefficient value may occur because of the lower viscosity, which can cause an increased curcumin release from this formulation [50]. However, the results of the in vitro permeation study showed that the diffusion rate of curcumin was greatly enhanced by using SNEDDS. It was found that in the case of the SNEDDS-containing formulations, the amount of drug release was at least two times higher compared with products without SNEDDS; it reached more than 30% within 4 h. This higher curcumin release from formulations containing SNEDDS may lead to higher bioavailability and thus a higher degree of efficacy [51].

To investigate the biocompatibility of our formulations, an in vitro cell viability study was performed on HaCaT cells. Possible toxic effects caused by the preparations on the cellular level were identified by MTT assay, where the live cells reduced yellow soluble MTT dye to insoluble formazan crystals. None of the compositions caused a cytotoxic effect on keratinocytes after treatment at the tested concentration (10 mg/mL). It was also found that creams caused higher cell viability values compared to gels. After 10 min of UVB irradiation, cell viability was also determined in order to characterize the UV protective effect of curcumin-containing products. Liu and his research group found that at least a concentration of 5 µM curcumin is needed for the protection against photodamage, tested on human fibroblast cells. In the mentioned study, it was described that the pretreatment with curcumin in a concentration of 2 mg/mL significantly reduced lactate dehydrogenase (LDH) leakage, indicating that curcumin has cytoprotective effect against 300 mJ/cm^2^ acute dose of UVB radiation in HaCaT cells [52]. According to our result, it can be concluded that all treatments with SNEDDS-containing formulations resulted in acceptable levels of viability, as it was above 70% in these cases. Creams and gels without SNEDDS have not been shown to be UV protective, as the cell viability has decreased significantly despite the treatments. This was probably because the active ingredient could not be released to such a large extent from the formulations without SNEDDS, as it was also presented in the permeation study.

Li et al. found that the UV protective effect of curcumin is related to the inhibition of intracellular ROS production [52]. Since cells use a natural antioxidant system to protect against high levels of ROS, we also measured SOD activity in pre- and post-treated HaCaT cells [53]. Our results demonstrated that treatments with formulations containing curcumin incorporated into SNEDDS resulted in significantly higher enzyme activity measured after UVB radiation. However, there was a smaller difference between the preparations with and without SNEDDS during the post-treatment. According to our findings, the effectiveness of creams and gels was significantly higher during the pretreatment. The results obtained were also supported by a study executed by Deng et al., who showed that curcumin (5 µM) pretreatment elevated SOD activity in HaCaT cells, measured after UVA (20 J/cm^2^) or UVB (57 mJ/cm^2^) irradiation compared with the control group [54]. With the help of the DPPH test, the direct antioxidant capacity of our SNEDDS-containing formulations was also proved. However, the difference between the radical scavenging activity of the compositions with and without SNEDDS was not as significant as when examining the indirect antioxidant effect on cells. The reason for this may be that the use of SNEDDS helped the active ingredient to enter the keratinocytes, and thus the development of the effect. Moreover, SNEDDS could also increase the solubility and decrease the biodegradation of lipophilic drugs, such as curcumin [55,56].

The significant anti-inflammatory activity of curcumin has attracted a lot of researchers’ interests and is one of the natural compounds with the greatest potential in the treatment of diseases [57]. This study demonstrated that treatments with creams and gels containing curcumin incorporated in SNEDDS significantly reduced the level of TNF-α and IL-1β investigated on the HaCaT cell line. A more significant decrease in TNF-α level was obtained in the case of the preparations containing the Pemulen TR-1 gel forming agent and Sedefos 75 emulsifier, so these compositions have been shown to be more effective in terms of anti-inflammatory effect in comparison with the Carbopol 974P gel forming agent and with the Tefose 63 emulsifier.

Curcumin has significant anti-inflammatory potential. Several preclinical and clinical research studies have investigated its effect on inflammatory diseases, as it is able to reduce the levels of the inflammatory mediators. As our results showed in the in vitro experiments, the SNEDDS-containing gel with Pemulen base and the SNEDDS-containing cream with Sedefos was selected for the in vivo experiments. The selected compositions were evaluated for their in vivo anti-inflammatory effects by carrageenan-induced paw edema in rats. The applied protocol has often used to evaluate the anti-inflammatory effect of the active ingredients. The development of carrageenan-induced edema is a two-phase phenomenon. The first phase (1 h after carrageenan injection) is mainly mediated by histamine, serotonin, and bradykinins, while the second phase of the inflammatory response (3 h after carrageenan injection) involves the release of various cytokines, such as IL-1β, IL-6, IL-10, and TNF-α [58,59]. In this study, the treatment with the curcumin + SNEDDS-containing formulations at the doses of 1–1 g displayed significant inhibition of paw edema in rats 4 h after carrageenan injection. We found that the anti-inflammatory effect of curcumin-containing cream and gel exceeded the anti-inflammatory effect of preparations containing diclofenac. According to the scientific literature, curcumin is able to reduce paw edema by the diminution of vascular permeability and reduction in leukocyte migration to the site of inflammation [60]. In a study conducted by Boarescu et al., better results were obtained in the anti-inflammatory test when curcumin was incorporated into nanoparticles. This was explained by the fact that the use of nanocarriers increased the tissue distribution of curcumin. According to their results, curcumin nanoparticles have also been shown to exert antioxidant effects in acute inflammation [61]. However, in the aforementioned studies, the results obtained during the oral administration of curcumin are presented. There are no data available in the literature on external, in vivo testing of nanoparticles containing curcumin.

Finally, the effect of the selected formulations upon UVB-induced skin inflammation was determined. There are very few studies on the topical application of curcumin and its in vivo anti-inflammatory effects. The in vivo anti-inflammatory effect of the topically used curcumin nanoparticles has been investigated by Zhang et al. on mouse suffering from osteoarthritis. They have described that topical treatment led to the reduced expression of pro-inflammatory mediators [62]. The effect of curcumin on UV-induced inflammation has never been investigated before. In our study, rats were exposed to UV radiation for 30 min after pretreatment. Six hours after irradiation, the level of TNF-α and IL-1β was determined from the tissue lysates. Based on our results, gel and cream formulations containing curcumin incorporated into SNEDDS showed significantly lower cytokine levels compared to the same formulations without active ingredient. It was also found that the amount of TNF-α decreased more significantly as a result of the treatment with the curcumin + SNEDDS-containing gel and cream.

Our work points out the importance of using nanoscale drug delivery systems for active ingredients with unfavorable bioavailability and solubility. As it was described earlier, previous studies have emphasized that the effectiveness of curcumin is greatly influenced by the type of the drug delivery system. The results of our in vitro and in vivo experiments have suggested that the addition of curcumin into SNEDDS can be a favorable solution to increase the water-solubility and the bioavailability. Further in vivo experiments and clinical trials are necessary to prove the relevance of the topical application of curcumin-loaded SNEDDS.

## 5. Conclusions

Taken together, the results suggest that curcumin incorporated into self-nanoemulsifying drug delivery systems can be a promising alternative option in the treatment of inflammatory skin diseases. The present study provides the first evidence that topically administered curcumin has significant efficacy in carrageenan-induced rat paw edema and in UV-induced inflammation as well. The anti-inflammatory and antioxidant effects, combined with the lack of toxicity, render curcumin a valuable candidate for further investigation as an agent for the treatment of various inflammatory skin diseases.

## Figures and Tables

**Figure 1 pharmaceutics-15-02054-f001:**
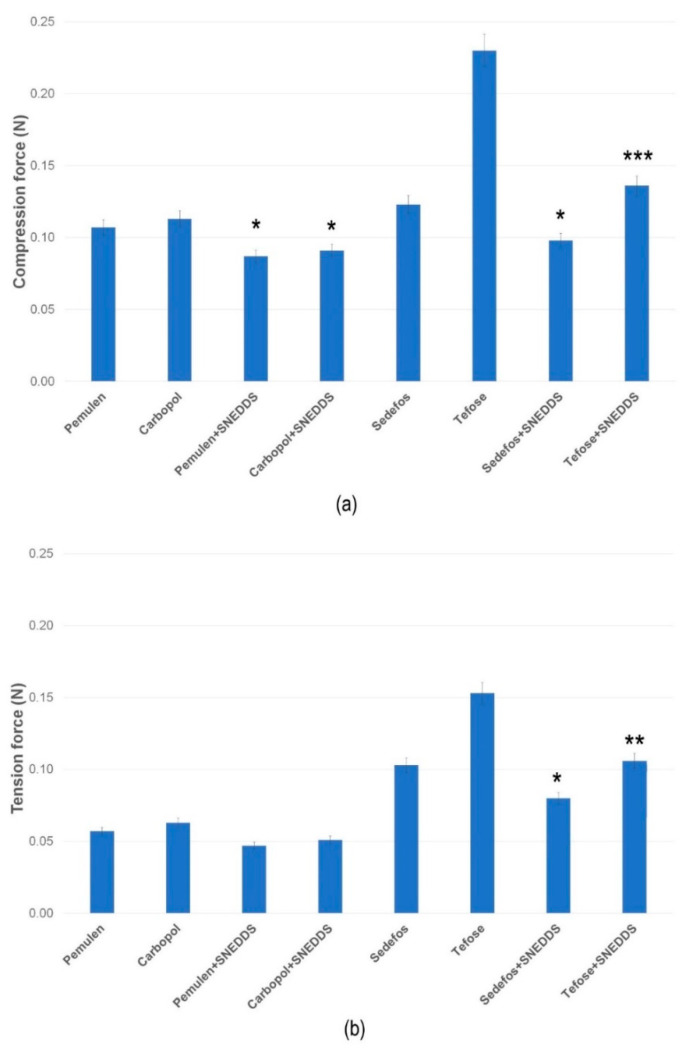
Texture analysis of the creams and gels at 25 °C, determined as compression (**a**) and tension force (**b**). Data are expressed as means ± S.D., *n* = 5. *, **, and *** indicate statistically significant differences between the formulations with and without SNEDDS at *p* < 0.05, *p* < 0.01, and *p* < 0.001.

**Figure 2 pharmaceutics-15-02054-f002:**
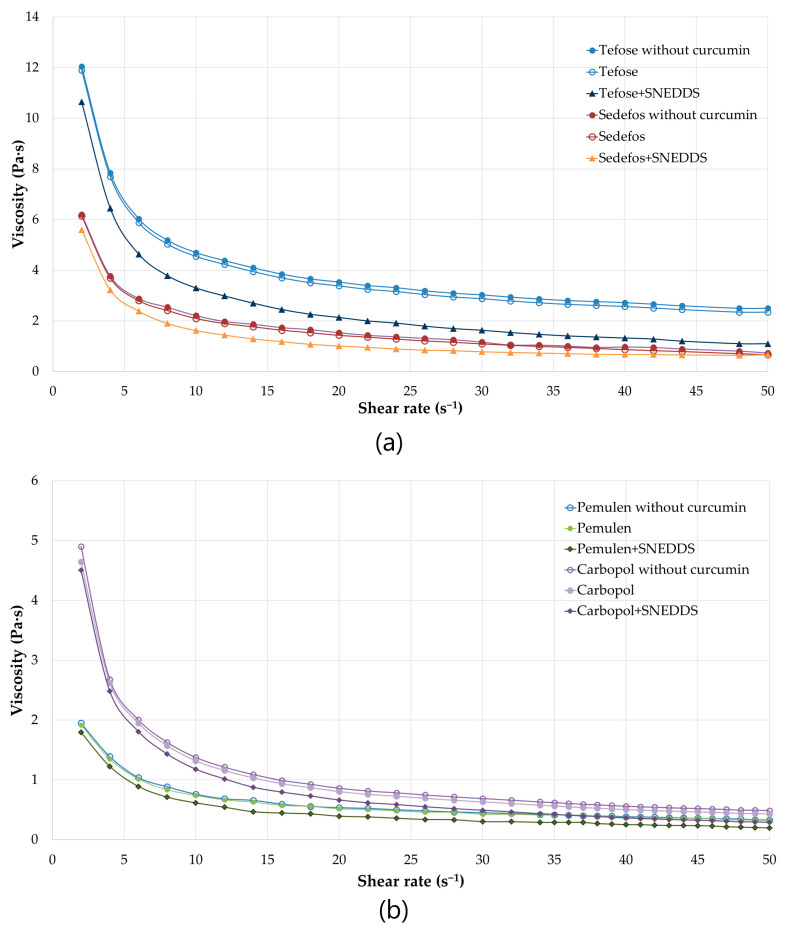
Viscosity (Pa·s) flow curves of the formulated creams (**a**) and gels (**b**) with or without curcumin as a function of controlled shear rate (s^−1^). Shear rate was increased from 2 to 50 s^−1^. Each data point represents the mean ± S.D., *n* = 3.

**Figure 3 pharmaceutics-15-02054-f003:**
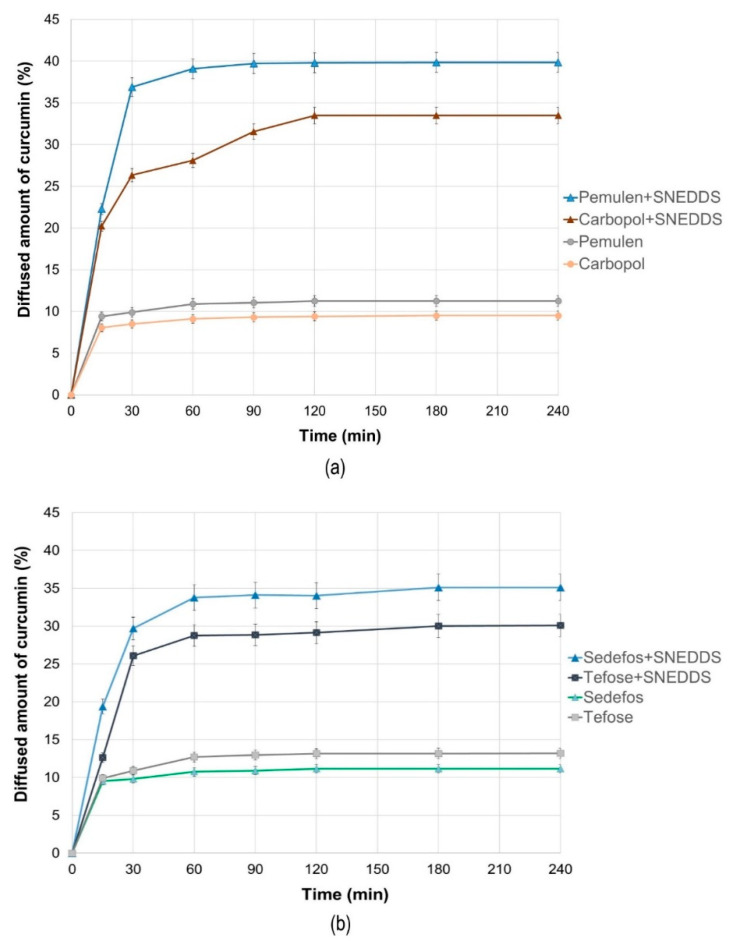
The in vitro diffusion profiles of curcumin from the gels (**a**) and creams (**b**) with and without SNEDDS. Bars represent the mean ± S.D., *n* = 6.

**Figure 4 pharmaceutics-15-02054-f004:**
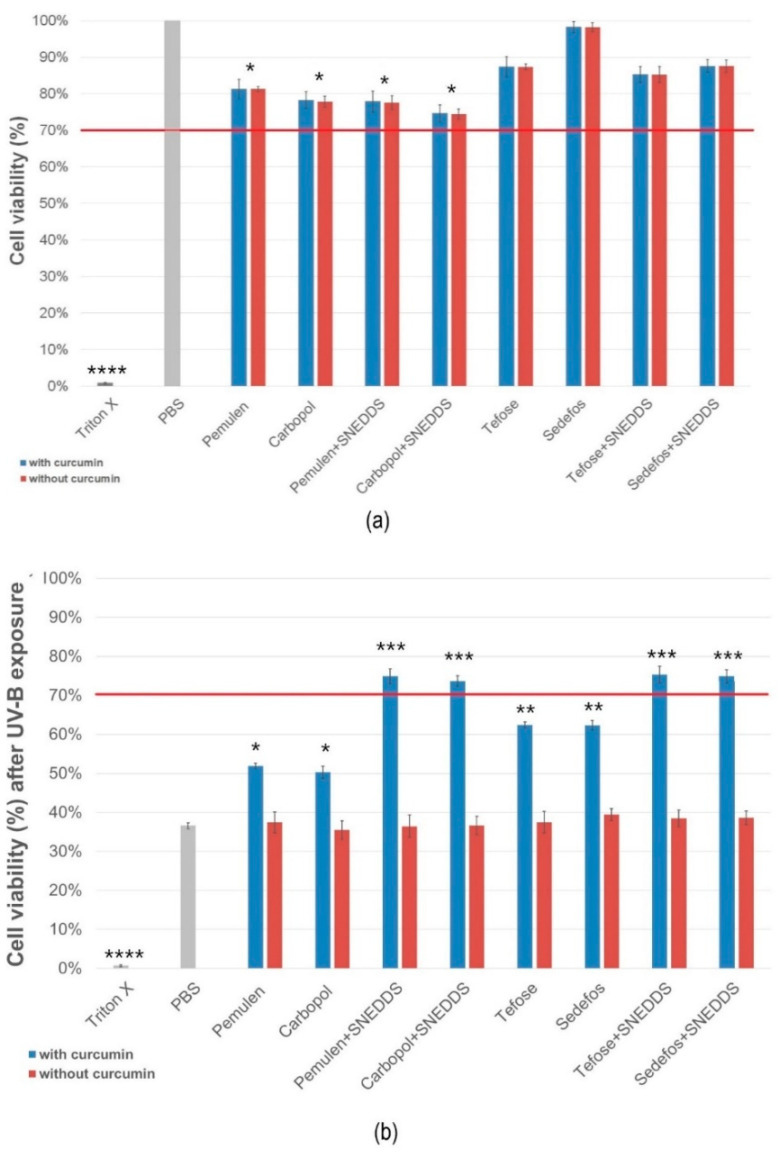
HaCaT cell viability after incubation of compositions (**a**) and viability after UVB radiation (**b**). Preparations resulting in cell viability above the red line (70%) are considered biocompatible according to the ISO 10993-5 recommendation. Data represent the mean ± S.D., *n* = 12. *, **, *** and **** indicate statistically significant differences between the samples and the PBS at *p* < 0.05, *p* < 0.01, *p* < 0.001 and *p* < 0.0001.

**Figure 5 pharmaceutics-15-02054-f005:**
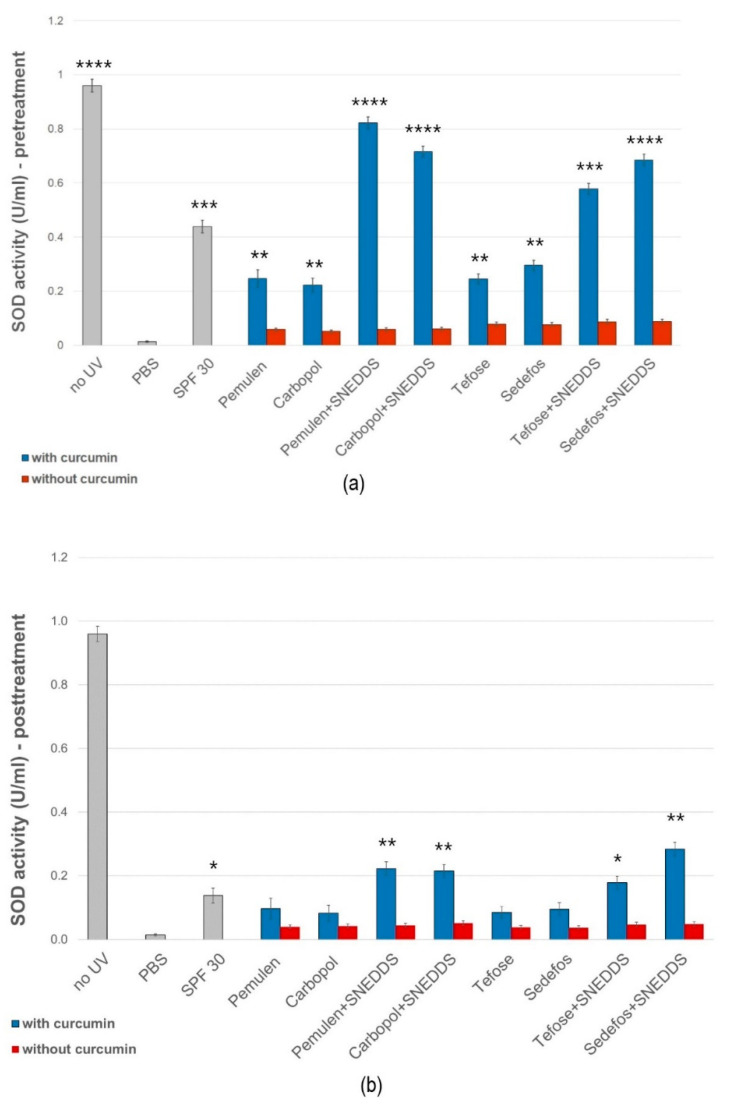
Effects of pre- (**a**) and post-treatment (**b**) with topical formulations with or without SNEDDS on SOD enzyme activity in HaCaT cells exposed to UVB irradiation. UVB irradiated cells pretreated with PBS were used as negative control. Data are expressed as the mean ± S.D. *n* = 10. *, **, ***, and **** indicate significant differences between the samples and the PBS at *p* < 0.05, *p* < 0.01, *p* < 0.001, and *p* < 0.0001.

**Figure 6 pharmaceutics-15-02054-f006:**
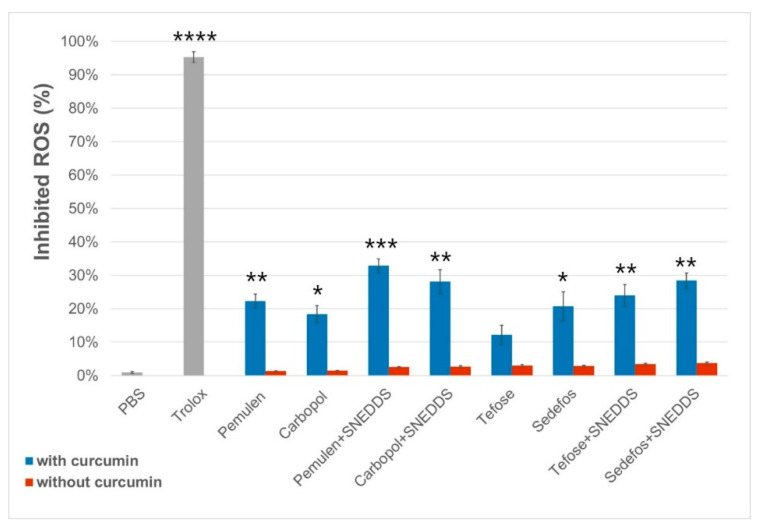
Radical scavenging activity of the formulations with and without curcumin. Each data point represents mean ± SD, *n* = 5. Significant differences between the formulations with or without curcumin are marked on the figure with *, **, *** and **** (*p* < 0.05, *p* < 0.01, *p* < 0.001 and *p* < 0.0001).

**Figure 7 pharmaceutics-15-02054-f007:**
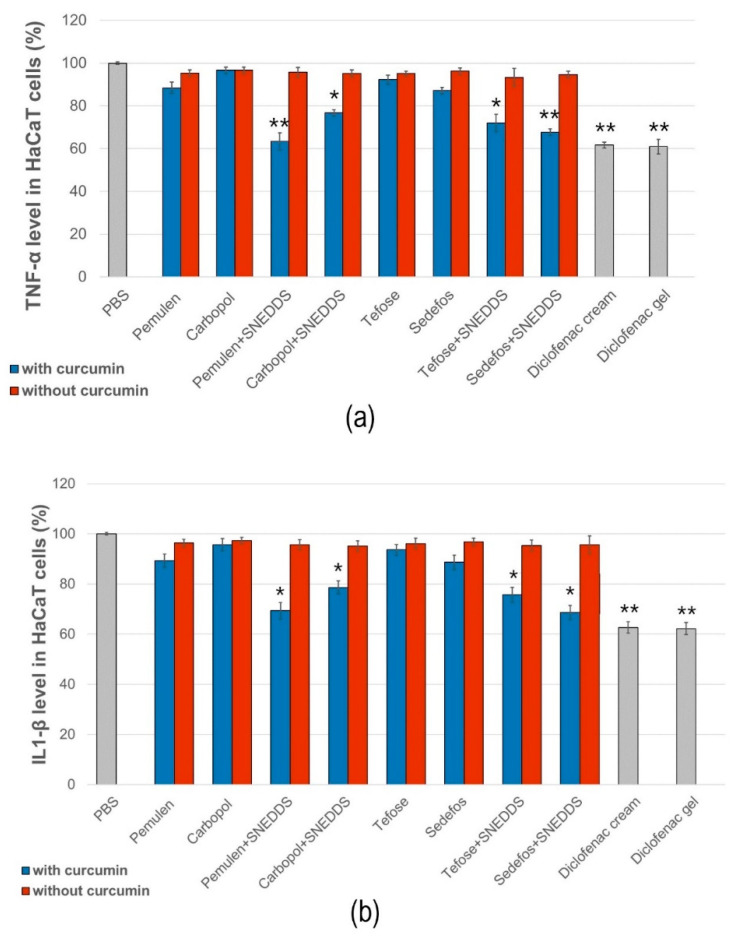
Human TNF-α (**a**) and IL-1β (**b**) ELISA tests on HaCaT cell line. Each data point represents means ± SD, *n* = 6. The * and ** (*p* < 0.05 and *p* < 0.01) show significant differences between the samples and the PBS.

**Figure 8 pharmaceutics-15-02054-f008:**
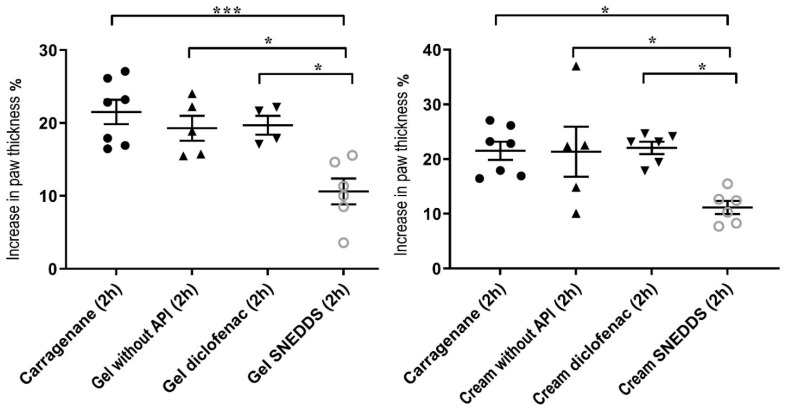
Effect of the treatment of the selected gel and cream compositions on the carrageenan-induced paw edema in the rat. Each composition was investigated on six animals. Results were expressed as means ± SD. The * and *** (*p* < 0.05 and *p* < 0.001) indicate significant differences.

**Figure 9 pharmaceutics-15-02054-f009:**
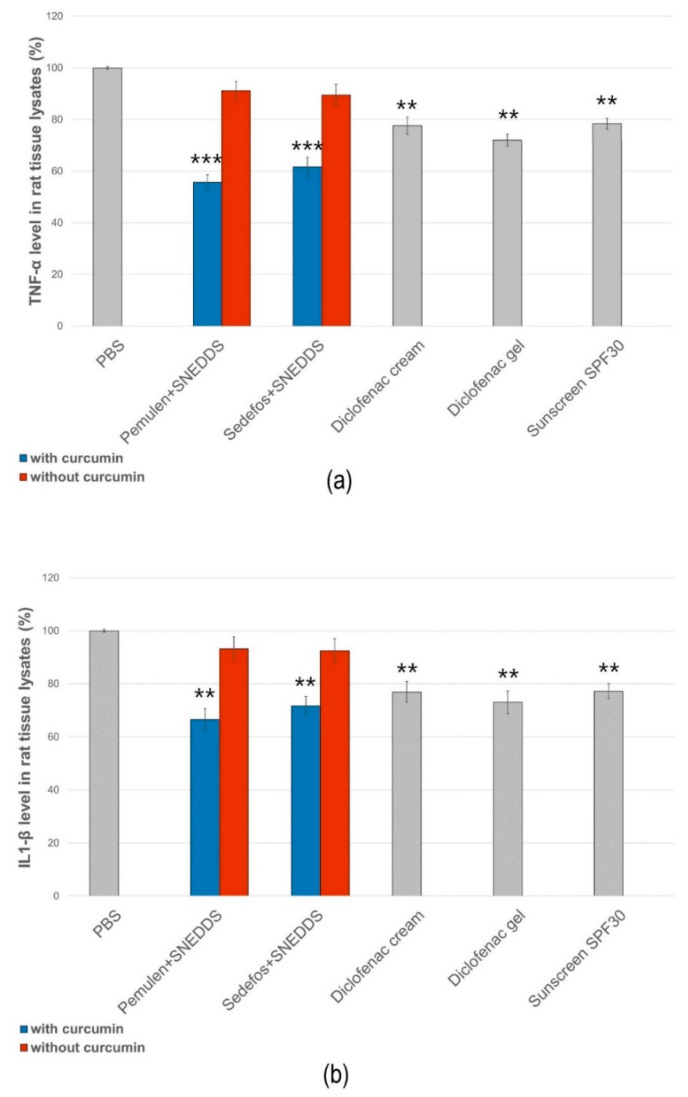
Effect of the treatment of the selected gel and cream compositions on UVB-induced skin inflammation. Each composition was investigated on six animals. The ** and *** (*p* < 0.01 and *p* < 0.001) show significant differences between the samples and the PBS.

**Table 1 pharmaceutics-15-02054-t001:** The composition of the formulated SNEDDS.

Transcutol P	Labrasol	Isopropyl Myristate (IPM)	Curcumin
37.5 g	37.5 g	15.0 g	10.0 g

**Table 2 pharmaceutics-15-02054-t002:** The compositions of creams containing curcumin or curcumin incorporated into SNEDDS.

	Tefose	Sedefos	Tefose + SNEDDS	Sedefos + SNEDDS
Curcumin	2 g	2 g	-	-
SNEDDS	-	-	20 g	20 g
Tefose 63	3 g	-	3 g	-
Sedefos 75	-	3 g	-	3 g
Cetylstearyl alcohol	4.5 g	4.5 g	4.5 g	4.5 g
Stearic acid	10 g	10 g	10 g	10 g
IPM	5 g	5 g	5 g	5 g
Propylene glycol	5 g	5 g	5 g	5 g
Distilled water	67.5 g	67.5 g	49.5 g	49.5 g

**Table 3 pharmaceutics-15-02054-t003:** The compositions of gels containing curcumin or curcumin incorporated into SNEDDS.

	Carbopol	Pemulen	Carbopol + SNEDDS	Pemulen + SNEDDS
Curcumin	2 g	2 g	-	-
SNEDDS	-	-	20 g	20 g
Carbopol 974P	0.6 g	-	0.6 g	-
Pemulen TR-1	-	0.6 g	-	0.6 g
Triethanolamine	1 g	1 g	1 g	1 g
Distilled water	96.4 g	96.4 g	78.4 g	78.4 g

**Table 4 pharmaceutics-15-02054-t004:** The pH values of curcumin-containing creams and gels immediately after formulation and after 30 and 60 days of storage at 21 °C. Values represent mean ± standard deviation (S.D.), *n* = 3.

Composition	pH Value ± SD		
	Immediately after Formulation	30 Days	60 Days
Pemulen	5.91 ± 0.05	5.90 ± 0.04	5.90 ± 0.05
Pemulen + SNEDDS	6.03 ± 0.03	6.04 ± 0.05	6.04 ± 0.05
Carbopol	5.88 ± 0.03	5.87 ± 0.04	5.86 ± 0.03
Carbopol + SNEDDS	5.95 ± 0.04	5.94 ± 0.03	5.94 ± 0.04
Tefose	5.72 ± 0.02	5.72 ± 0.04	5.71 ± 0.04
Tefose +SNEDDS	5.84 ± 0.04	5.84 ± 0.03	5.83 ± 0.05
Sedefos	5.77 ± 0.03	5.75 ± 0.04	5.76 ± 0.03
Sedefos + SNEDDS	5.91 ± 0.02	5.90 ± 0.04	5.90 ± 0.04

**Table 5 pharmaceutics-15-02054-t005:** Spreadability values of the different compositions at room temperature. Values represent mean ± standard deviation (S.D.), *n* = 3.

Composition	Spreadability (cm/5 min)
Tefose without curcumin	3.4 ± 0.10
Tefose	3.4 ± 0.05
Tefose + SNEDDS	3.9 ± 0.12
Sedefos without curcumin	3.6 ± 0.10
Sedefos	3.6 ± 0.12
Sedefos + SNEDDS	4.1 ± 0.14
Pemulen without curcumin	5.2 ± 0.16
Pemulen	5.2 ± 0.15
Pemulen + SNEDDS	5.8 ± 0.22
Carbopol without curcumin	5.1 ± 0.16
Carbopol	5.1 ± 0.15
Carbopol + SNEDDS	5.6 ± 0.20

**Table 6 pharmaceutics-15-02054-t006:** Release rate and the diffusion coefficient values ± S.D., *n* = 6. Significant differences between the compositions with and without SNEDDS are signed with * (*p* < 0.05).

Composition	Release Rate (k) (µg/cm^2^ × min½)	Diffusion Coefficient (D) (cm^2^/min)
Pemulen	12.413 ± 1.018	0.01593 ± 0.00112
Carbopol	10.328 ± 1.321	0.01110 ± 0.00098
Pemulen + SNEDDS	50.683 ± 3.023	0.20479 ± 0.01543 *
Carbopol + SNEDDS	43.872 ± 2.162	0.10588 ± 0.01182 *
Tefose	15.561 ± 1.666	0.02159 ± 0.00177
Sedefos	12.208 ± 1.398	0.01550 ± 0.00109
Tefose + SNEDDS	41.339 ± 2.443	0.11088 ± 0.00987 *
Sedefos + SNEDDS	45.538 ± 3.255	0.15290 ± 0.00876 *

**Table 7 pharmaceutics-15-02054-t007:** The comparison of diffusion profiles of the formulations with or without SNEDDS by the calculation of the difference (*f*_1_) and the similarity factor (*f*_2_).

Pairwise Comparison	*f*_1_ ^a^	*f*_2_ ^b^
Pemulen vs. Pemulen + SNEDDS	70.88	28.71
Carbopol vs. Carbopol + SNEDDS	69.35	33.98
Tefose vs. Tefose + SNEDDS	53.67	41.16
Sedefos vs. Sedefos + SNEDDS	66.36	33.37

^a^ *f*_1_ values of the difference factor calculation; ^b^ *f*_2_ values of the similarity factor calculation.

## Data Availability

The data that support the findings of this study are available from the corresponding author (jozsa.liza@euipar.unideb.hu) with the permission of the head of the department, upon reasonable request.

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
