# Peer review of "In Vitro and In Vivo Efficacy of Topical Dosage Forms Containing Self-Nanoemulsifying Drug Delivery System Loaded with Curcumin"

_pharmaceutics, 2023, doi:10.3390/pharmaceutics15082054_

Round 1
Reviewer 1 Report
The manuscript is well written. It is suggested to be accepted.
Author Response
Dear Reviewer 1,
I hereby submit our manuscript “In vitro and in vivo efficacy of topical dosage forms containing self-nanoemulsifying drug delivery system loaded with curcumin”, by Gréta Frei et al. which has been modified according to the suggestion of the other Reviewers. Thank you for your kind review.
Sincerely,
Liza Józsa
corresponding author

Reviewer 2 Report
Frei et al. report creams and gels containing curcumin prepared from penetration enhancing surfactants and gelling agents as potential formulations for topical applications. The various in vitro and in vivo studies demonstrated the effectiveness of formulations for the treatment of inflammatory skin diseases. This study is very interesting and can result in significant contribution in the field of biomedicine and drug delivery. Nevertheless, the following comments and suggestions must be addressed before the article can be accepted for publication in Pharmaceutics:
1. The full names of DNA and SNEDDS in the first and last paragraph of introduction section must be provided before they can be abbreviated, respectively.
2. The citation number must be before the full stop of each sentence. This must be corrected throughout the manuscript.
3. In line 62, Curcuma longa (C. longa) should be written in italics.
4. In line 516, the word ‘’he’’ should be corrected to ‘’the’’.
5. Check line 595, is curcumin contains active ingredients with antioxidant and anti-inflammatory potential or is it an active ingredient with antioxidant and anti-inflammatory potential? It will be understood if it is C. longa that contains the active ingredients with antioxidant and anti-inflammatory efficacy, curcumin included as an active agent.
6. In line 602, is it required to write the full name of SNEDDS? Since the authors have been writing the abbreviation in the previous sections. Also, ROS in line 643.
7. This research manuscript lacks the spectroscopic characterizations of formulations, e.g., FTIR.
8. The viscosity and spreadability tests of formulations with and without curcumin, especially gels, supposed to be conducted and compared because they are very important aspects in topical applications.
Author Response
Dear Reviewer 2,
I hereby submit our modified manuscript “In vitro and in vivo efficacy of topical dosage forms containing self-nanoemulsifying drug delivery system loaded with curcumin”, by Gréta Frei et al. Thank you for your constructive comments and suggestions, according to them, the following changes were made during revision (corrections related to the Review 2 are marked with yellow in the manuscript):
- The full names of DNA and SNEDDS in the first and last paragraph of the introduction has been added. Thank you for your comment and sorry for the mistake.
- The position of the citation numbers has been corrected. It has been added before the full stop of each sentence.
- In line 62 the Curcuma longa (C. longa) has been corrected to italics.
- In line 541 the word ‘’he’’ has been corrected to ‘’the’’.
- In line 620 the mentioned sentence has been corrected: The external use of curcumin is not so widespread, although it is a valuable active ingredient with antioxidant and anti-inflammatory potential.
- In line 626 and in line 667 the full names have been corrected to the abbreviations (SNEDDS and ROS), according to the suggestions.
- FT-IR measurement of the curcumin and the SNEDDS has been carried out, according to the suggestion. Unfortunately, spectrophotometric evaluation of creams and gels could not be performed due to the large number of components in them.
The infrared spectra of pure curcumin, curcumin-free, and curcumin-containing SNEEDS were obtained with a JASCO FT-IR 4600 (ABL&E-JASCO, Budapest, Hungary) equipped with a Zn/Se ATR PRO ONE Single-Reflection ATR accessory. All samples were placed directly on the crystal of the ATR device. The scanning was performed 32 times in the wavelength range 500-4000 cm-1 with a resolution of 1 cm-1, and the correction for background CO2, H2O, and automated baseline correction and smoothing was performed using the built-in method of the software. The spectra were evaluated as described in our previous studies [1].
Figure 1. FTIR spectra of curcumin, curcumin-free, and curcumin-containing SNEEDS. (Please see the attachment)
FTIR spectroscopy measurements were used to study the interaction between curcumin and the tensides forming the basis of SNEEDS. The spectral of curcumin shows the characteristic peaks in the range of 1500-1200 in Figure 1. Comparing SNEEDS without and with curcumin, the characteristic peaks of curcumin in the spectrum can be observed. A shift is observed for peaks of curcumin 1505, 1273, and 1203. In the SNEEDS samples, new peaks appear at 1513, 1283, and 1206, suggesting a chemical interaction. The aromatic stretching, carbonyl bond vibration observed at 1512 cm-1. Intense band at 1273 cm-1 attributed to the bending vibration of the phenolic band [2]. The shifting of the peaks is assumed to be due to the secondary interaction of the free C=O and C-OH groups of the surfactants.
Since we only measured the SNEDDS itself (not the creams and gels), we thought that it didn't quite fit into the topic of the article, so it was left out from the article.
- The viscosity and the spreadability of the formulations with and without curcumin have been investigated, according to the suggestion. The methods and the results of these measurements have been added to the article (Chapter 2.5. and 3.3.)
[1] H. P. Le Khanh et al., “Effect of Molecular Weight on the Dissolution Profiles of PEG Solid Dispersions Containing Ketoprofen,” Polymers (Basel)., vol. 15, no. 7, 2023, doi: 10.3390/polym15071758.
[2] E. H. Ismail, D. Y. Sabry, H. Mahdy, and M. M. H. Khalil, “Synthesis and Characterization of some Ternary Metal Complexes of Curcumin with 1,10-phenanthroline and their Anticancer Applications,” J. Sci. Res., vol. 6, no. 3, pp. 509–519, Aug. 2014, doi: 10.3329/jsr.v6i3.18750.
We hope that our improved manuscript is worth publication.
Waiting for your kind response.
Sincerely,
Liza Józsa
corresponding author

Reviewer 3 Report
1. The sentence in lines 49-50 does not sound the best, read it and try to rephrase it.
Topical dosage forms, like ointments, creams and gels containing herbal active ingredients are also very common among these preparations.
2. Mix drug, herbal active ingredients and curcumin
3. You mix the cosmetic product and the medicine. A cosmetic product does not heal! It is used for care, improvement of appearance, cleaning, washing, protection of the skin... It does not cure and if your preparations only protect the skin from the sun's rays, they are cosmetic products, not drugs at all and the term drug cannot be used, but if it is a drug then it cannot be a cosmetic product. Unfortunately, you are mixing up those terms in the introduction. Read. After you read it, you can see that you want them to have healing properties as well. Be a little more precise in your expression, it's a little unclear what we expect. A little medicine, a little cosmetic product, a little sun protection, and the next sentence anti-inflammatory effect and cancer. Make an introduction that this is a substance used in both the cosmetic and pharmaceutical industries. For a cosmetic purpose... for a pharmaceutical purpose... to make it clearer to the reader... Mechanism... and you also link to my next comment excellent.
4. Lines around 88-89 you are not precise, biodistribution where... is there any data on whether it is permeation or penetration through the skin, where the site of its action is the epidermis, dermis, or does it go to the dermis and systemic circulation. This part is very interesting, try to clarify a little, because the mechanism of action will probably be clearer later, but let's know here where in the skin its action is expected, which mechanism.
5. And in order to bioavailability. Are we talking about a systemic effect? Is the bioavailability improved, passing through the skin, into the skin...
6. When introducing some abbreviations, have them in the objective, but also in the entire work, please use the full title.
7. Table 1 No abbreviation IPM. How do the component concentrations compare to other similar studies?
8. Why exactly this temperature for preparation, I see that you emphasize it, is there any reason?
9. Please use the word self-nanoemulsifying drug delivery system instead of self-nanoemulsifying system in the title of table 1 and the entire paper. In order for the abbreviation to be correct and for us to be uniform. You can see what you have in the second SNEDDS table from the first one, and you didn't call it that.
10. What is the usual concentration of IPM in skin preparations? Interesting formulation of the cream, you also have the co-emulsifier cetostearyl alcohol. The question is, does it have a base component so that with stearic acid, stearin cream, modified stearin cream is formed? Or is stearic acid here just a consistency modifier and a component of the fat phase of the cream.
11. Did the fat phase melt when making the cream? This is quite a low temperature for stearic acid, it melts at around 80 degrees. Please check.
12. Please check whether triethanolamine solution or concentrated was used to make the gels, for focused this is a large amount for me, I am afraid that the mixture will completely disintegrate due to the high base value. Check how much triethanolamine is added to neutralize the carbopol.
13. Perhaps my narrowest area, it is correct that you reference the paper, but it would be nice if you could state somewhere what the drop size of those SNEDDS and PDI was with and without curcumin.
14. Missing reference to lines 216-220
15. Row 234 phosphate buffer what pH value?
16. Is the procedure in chapter 2.7. according to some procedure or...? Explain why the skin test temperature for some tests is 37°C when the standard for skin preparations is 32°C. It would be nice to have references.
17. Units spaced from the number, everything except %
Minor editing of English language required
Author Response
Dear Reviewer 3,
I hereby submit our modified manuscript “In vitro and in vivo efficacy of topical dosage forms containing self-nanoemulsifying drug delivery system loaded with curcumin”, by Gréta Frei et al. Thank you for your constructive comments, according to them, the following changes were made during revision (corrections related to the Review 3 are marked with green in the manuscript):
- The mentioned sentence “Topical dosage forms, like ointments, creams and gels containing herbal active ingredients are also very common among these preparations.” has been rephrased as the following: In case of topical formulations many types of plant extracts are used, with a variety of chemical ingredients, for example, flavonoids, tannins, phenolics, aminoacids and vitamins, which influence the biological functions of the skin. [https://doi.org/10.3390/cosmetics8040106]
- The text of the article has been reworded, possible misunderstandings have been corrected, the differences between the drug and the herbal active ingredient have been clarified.
- The part of the introduction about curcumin and its areas of use (cosmetic/pharmaceutical industry) has been separated and clarified, according to the suggestion. The mechanism of action has been described in more detail.
- When curcumin is incorporated into a drug delivery system, for example to a nanoparticle, it is able to penetrate even into the dermis. According to the literature, nanocarrier systems make it possible to enhance the permeation of curcumin through the stratum corneum. Ex vivo skin permeation and deposition studies have been confirmed that the small size of these nanoparticles allows them to closely interact with the stratum corneum cells and improve permeation and accumulation in the epidermal and dermal region [doi:10.3390/molecules17055972.]. The site of action in the skin has also been described in the introduction, according to the suggestion.
- As curcumin is a lipophilic compound revealing log P = 3.29 and relatively low molecular mass [doi: 10.3390/biom9020056], it may be expected that transdermal products can be advantageous in terms of drug bioavailability. However, in vitro skin permeation studies performed by Fang et al. [doi: 10.1211/002235703765344496] indicate that curcumin flux without any additional percutaneous absorption enhancers is very low. The application of a proper drug delivery system allows for transporting the active ingredient through stratum corneum, the most important skin barrier, into the deeper skin layers. [doi: 10.3390/app13137809] Even after consuming high amounts of conventional curcumin, very low levels of plasma curcumin were detected, as it eliminates rapidly. [doi: 10.1021/acsomega.2c07326] The systemic absorption of the topical formulations is undesirable, the absorption of the topically used curcumin would not occur, which may be regarded as an advantage. Goncalves et al. has analyzed the distribution of curcuminoid pigments in the skin, and they have concluded that a significant amount of curcumin was delivered to the epidermis and dermis. Furthermore, it was found that the curcumin was delivered into the basal layer of the epidermis and the upper layer of the dermis. These are usually the target skin layers for active ingredient delivery in case of topical formulations. [doi: 10.1590/S1984-82502014000400024]
- Thank you for your suggestions. The abbreviations and their full title have been corrected.
- The surfactant-oil ratio in SNEDDS is quite variable. In a similar study the surfactants:IPM ratio was for example 1:9, when Paclitaxel was incorporated to SNEDDS. [doi: 10.1155/2016/3642418] The amount of the oil phase can be increased or decreased depending on the solubility of the active ingredient. In a study where a microemulsion system has been formulated the ratio of Tween80:Span20:lauric acid:IPM:curcumin was 33.3:1.6:1:5:1.3 [doi: 10.1155/2017/5205471] I did not find any publication where IPM was used in the formulation of curcumin-containing SNEDDS. According to our previous experiment, a 5:1 surfactant-oil ratio proved to be the most favorable.
- According to our previous investigations the applied tensides are remaining stable in this temperature, heating should be avoided. It was emphasized just to clarify that no melting is needed in order to mix them.
- Thank you for your kind remark, the word self-nanoemulsifying drug delivery system has been used instead of self-nanoemulsifying system in the title of Table 1 and it has been corrected in the entire paper.
- The usual concentration of IPM in creams and gels is 1-20%. In our SNEDDS containing creams the concentration of the IPM is 8%, in creams without SNEDDS it is 5%, while in the SNEDDS containing gels it is 3%. According to the literature these concentrations are safe for the skin. [doi: 10.1016/S2352-4642(22)00146-8] The purpose of using stearic acid was only to modify the consistency, that is why it was added to the fat phase of the creams.
- The temperature applied in the formulation of creams was corrected to 80 °C. Thank you for your comment and sorry for the mistake.
- In our gels the amount of the triethanolamine was 1%, which is considered an average amount according to the literature. [doi: 10.4103/2231-4040.93564, doi: 10.4103/2231-4040.93564] I have also found a study in which 4% triethanolamine was used for the preparation of a Carbopol gel. [doi: 10.24071/jpsc.002562] Our mixtures were not disintegrated, they remained stable.
- The droplet size and the polydispersity index value has been added to the chapter 2.2.
- In lines 235-239 the missing reference was added: Vasvári G, Haimhoffer Á, Horváth L, et al. Development and Characterisation of Gastroretentive Solid Dosage Form Based on Melt Foaming. AAPS PharmSciTech. 2019;20(7):290. Published 2019 Aug 19. doi:10.1208/s12249-019-1500-2
- In line 253 the pH of the phosphate buffered saline (PBS) solution has been added: It was pH 7.4.
- In case of the investigations which include cell lines the required temperature is 37 °C, because according to the official guidelines this incubation temperature is necessary for the cells to survive. [doi: 10.1016/B978-0-12-803077-6.00009-6, doi: 10.3390/pharmaceutics15041146] For example, Li and his research group has also tested the ultraviolet (UV) radiation protective effect of a natural extract on HaCaT cells with the help of the same method at 37 °C. [doi: 10.1016/j.jphotobiol.2018.12.001] This reference has been added to the article.
- The units has been spaced from the numbers in the whole article, expect %.
We hope that our improved manuscript is worth publication.
Waiting for your kind response.
Sincerely,
Liza Józsa
corresponding author

Round 2
Reviewer 3 Report
Acceptic in present form.